# Tracking of Thermal, Physicochemical, and Biological Parameters of a Long-Term Stored Honey Artificially Adulterated with Sugar Syrups

**DOI:** 10.3390/molecules28041736

**Published:** 2023-02-11

**Authors:** Monika Tomczyk, Anna Czerniecka-Kubicka, Michał Miłek, Ewelina Sidor, Małgorzata Dżugan

**Affiliations:** 1Department of Chemistry and Food Toxicology, Institute of Food Technology and Nutrition, University of Rzeszow, Ćwiklińskiej 1a St., 35-601 Rzeszow, Poland; 2Department of Experimental and Clinical Pharmacology, Medical College of Rzeszow University, University of Rzeszow, 35-310 Rzeszow, Poland; 3Doctoral School, University of Rzeszow, Poland, Rejtana 16c, 35-959 Rzeszow, Poland

**Keywords:** honey, adulteration, sugar syrup, temperature of glass transition, sucrose, fructose, glucose, viscosity

## Abstract

The growing phenomenon of honey adulteration prompts the search for simple methods to confirm the authenticity of honey. The aim of the study was to evaluate the changes in thermal characteristics, physicochemical parameters, antioxidant and enzymatic activity of honey subjected to artificial adulteration. Two series of products were prepared with the use of two different sugar syrups with an increasing dosage of adulterant (0 to 30%). After 24 months of storage, the quality of adulterated samples (partially crystallized) was assessed in comparison to the control honey (solid). Used adulteration changed physicochemical parameters and reduced antioxidant and enzymatic activity of honey (*p* < 0.05). The admixture of syrup and invert (*p* < 0.05) reduced the viscosity of liquid phase of delaminated honey in a dose-dependent manner. In the study, artificially adulterated honeys were controlled using the standard differential scanning calorimetry, DSC. In all adulterated honeys, a specific glass transition, TG, was observed in the range of 34–38.05 °C, which was not observed for control honey and pure adulterants. Moreover, the additional T_g_s were observed in a wide range from −19.5 °C to 4.10 °C for honeys adulterated by syrup only. In turn, the T_g_ in range of 50.4–57.6 °C was observed only for the honeys adulterated by invert. These specific T_g_ seem to be useful to detect honey adulteration and to identify the kind of adulterant used.

## 1. Introduction

Good quality honey is a desirable product on the food market. The decline in the population of bees of unknown etiology and climate change has a negative impact on the productivity of beekeeping farms and as a result it limits the honey supply. This situation creates the circumstances for the increasingly common phenomenon of honey counterfeiting [1]. Among the food products subject to adulteration, honey still occupies a high position, next to products such as olive oil, saffron, coffee, grape wine, maple syrup or vanilla [2]. Honey adulteration generally takes two forms: modification of the composition, e.g., by the addition of foreign substances, as well as incorrect marking of the variety and origin of the honey. The addition of foreign substances can occur directly, as a result of bee feeding, or by mixing high-quality honey with cheaper, low-quality honey [3].

The most commonly added foreign substances are sugar syrups such as corn syrup or high-fructose corn syrup, inverted syrup, rice syrup, maple syrup [4,5,6,7,8], as well as sucrose [9] or fructose and glucose [10]. In addition to sugars, there are also additions of sulphite-ammonia caramel, which results in a dark color and thus a higher price of honey [1]. The adulteration of honey results in the loss of its valuable biological properties, although it does not reduce its sweetening properties. Mixing honey with imported honeys of poor quality may also introduce foreign pollen and result in the development of allergies.

There is a constant need to search for new analytical methods to detect honey adulteration. The most used are physicochemical [11,12], chromatographic [6,13,14,15] and spectroscopic [8,9,10,16] methods. The 13C/12C isotope ratio analysis is a method that allows for the detection of the addition of starch syrup, which allows to determine whether sugars from plants belonging to the C-3 pathway have been introduced [17]. A new approach is the use of molecular methods based on the detection of corn DNA traces and, as a result, adulteration with corn syrup [7].

The use of thermal analysis is a relatively new approach. It facilitates the detection of differences in the behavior of natural honey and honey doped with sugar syrups during analysis with the differential scanning calorimetry (DSC). The addition of water, sucrose, glucose and fructose introduced together with sugar syrups inside the honey sample has an impact on the change of thermal parameters, as compared to raw honey [18,19,20].

The aim of our research was to assess the possibility of using physicochemical parameters, biological activity and temperature of glass transition, in order to quickly identify honey adulteration with sugar syrup or invert. The study focused on self-prepared model systems “honey + adulterant” which were stored in real conditions for a long time before analysis.

## 2. Results and Discussion

### 2.1. Long-Term Stored Honey Behavior and Appearance

Honey is a bee product that is naturally changing from a liquid to a solid state. Our earlier studies examined the effect of various preservation methods on the quality of varietal honeys. It was observed that ultrasonically decrystallized honey retains its liquid form for longer than after traditional heat treatment, while retaining its valuable ingredients, and later re-crystallizes [21]. The rate of the crystallization process of honey depends on its density and the percentage of water content. The more saturated liquid honey is with sugars, the faster the crystallization process and thus the transition into a solid state. Therefore, the sugar content is a very important factor and its ratio to the other ingredients, especially water. More glucose results in faster crystallization of honey. During the crystallization of honey, the so-called delamination of honey into a solid and liquid phase often occurs [22].

The addition of sugar syrup into fresh liquid honey allowed us to obtain two series of homogeneous products: HS (honey syrup mix) and HI (honey invert mix), with a changed appearance (color and viscosity) compared to raw honey. Using an additive share that increased from 5 to 30% *w*/*w*, two groups of artificially adulterated honey were prepared: HS 5, HS 10, HS 20, HS 30 and HI 5, HI 10, HI 20, HI 30, for syrup and invert, respectively. During 24 months of storage at room temperature (18 ± 2 °C), the honey with additives partially crystallized, while the control honey crystallized completely. The adulterated samples showed a biphasic structure: the bottom was a solid phase (HSS and HIS for honey syrup and invert solids, respectively), and the top was a liquid phase (HSL and HIL for honey syrup liquid and honey invert liquid, respectively) and the phases ratio varied depending on the applied additive (Table 1).

One of the physical parameters of honey, mainly used to distinguish between pure and adulterated honey, is its color. According to the reports of Gebremariam and Brhane (2014) [23] and Gemeda and Negera (2017) [24], adulterated honey can be distinguished from real honey by visual color assessment. In the present study, the honey color was assessed instrumentally according to the Pfund scale (Table 1). The measurements showed that with the increase of the adulterant content, the Pfund values decreased proportionally. In the case of 30% syrup addition a drop in the Pfund value by as much as 26% was observed. Less visible decreases were observed in the case of the invert, which is related to the color of the pure adulterant.

### 2.2. Physicochemical Parameters

The commercial quality of honey is assessed according to the applicable regulations based on a set of physicochemical parameters, including, e.g., water content, sugar content, pH, conductivity, HMF (5-hydroxymethylfurfural) content. Exceeding the applicable limits may be the first signal of incorrect chemical composition of honey, which may be, among others, a consequence of adulteration. Honey adulteration caused by the addition of cane sugar, invert or sugar syrup can be reflected in many components, such as: changes in HMF content, sucrose, glucose to fructose ratio or diastase activity; as well as color, consistency, water content, ash content and the number of insoluble particles [25,26,27]. In control honey the water content was at the level of 17.9%, which is an average result, as according to the International Honey Commission (IHC) [28], the water content in honey should be within the range of 16.4–20%. The addition of both invert and sugar syrups gradually increased the water content in the prepared samples even up to 21% (in the case of HI 30 and HS 30 samples) (Table 1). Honey moisture determines the capability of honey to resist spoilage by yeast fermentation [28]. In terms of microbiological stability, water activity is a better parameter. The value below 0.6 completely stops yeast and fungal growth [26]. An elevated water activity measured for adulterated honey could negatively influence samples stability during storage, especially with the increase of temperature conditions. Our observation is in line with the study of Yilmaz et al. (2014) [29], where honey samples adulterated with fructose and saccharose syrup were evaluated.

In the pH values of the adulterated samples, an increase in the concentration of artificial sugar addition was noted, however, the differences were not found significant (*p* > 0.05) compared to control honey (4.5), which was still within the range typical for genuine honey (3.4–6.1) [30]. The changes in pH can indicate the fermentation process. Gebremariam and Brhane (2014) also observed that pH value increase, whereas free acidity decreases upon addition of commercial sugar [23].

The electrical conductivity of honey is closely related to the concentration of mineral salts, organic acids and proteins and it strongly depends on the honey variety [31]. It was found that an increasing addition of syrups caused a proportional decrease in specific conductivity (*p* < 0.05). The comparative study conducted by Yakubu et al. (2021) indicated that main value for a pure sample was 0.28 mS/cm, with a decrease to 0.17 mS/cm for mixture of honey and sugar (1:1 *w*/*w*) [32]. This change results from the low content of mineral salts, organic acids and proteins introduced with adulterants [33]. 

The conducted analysis clearly shows that the admixture of syrup and invert significantly reduces the viscosity of liquid phase of delaminated honey, measured at a temperature of 18 ± 2 °C. The control honey showed a viscosity close to the literature data, considering a close correlation of this parameter with temperature [34,35]. A linear correlation of a drop in viscosity with an increase of the percentage of syrup in the mixture was found (Figure 1). As the viscosity of the syrups used to prepare the adulterated samples was much lower compared to control honey (by about 97% lower) this tendency was easy to predict. A drop in viscosity was observed, at first for honey adulterated by more than 5% of the syrup [11,36]. Thus, obtained results can indicate that viscosity may serve as an indicator of the presence of adulterants.

The addition of syrup changed the sugar profile of adulterated honeys, which was tested by the high-performance thin-layer chromatography (HPTLC) method. Semi-quantitative analysis of the generated chromatograms allowed to calculate the fructose to glucose ratio (F/G ratio) in the tested samples (Figure 2). The F/G ratio gradually increased with an addition of syrups in the adulterated honey (from 0.62 to 1.0 for control and HIL 30 sample, respectively). It is commonly known that the honey sugar profile influences the process of honey crystallization, as well as that the time required for honey to crystallize depends mostly on the F/G ratio [37,38]. The significance of the botanical origin of honey on the sugar ratios: F/G and glucose to water was reported. Additionally, a lower F/G ratio and water content were correlated with a faster crystallization of the honey [37].

HMF, a product of acid inversion, was measured in all samples by spectrophotometric method. The results indicated that addition of inverted sugar or syrup increased the HMF level (*p* < 0.05), however the amounts were still within the maximum limit (40 mg/kg) regulated by IHC [28]. It was observed that favorable conditions for the HMF formation occurred in the liquid phase of delaminated honey, as compared to the solid phase. According to Tura and Seboka [39], HMF can be used as an index to detect the presence of invert syrups in honey. Freshly bottled honey should contain no HMF [40], however the HMF is primarily formed during heating or long storage of honey, therefore the validity of HMF as an adulterant indicator is questionable [26].

### 2.3. Bioactivity Indicators

The increasing addition of syrup in both cases decreased the antioxidant activity of honey in a dose-dependent manner, regardless of the research method used, however the observed changes were smaller in the case of invert (Table 2). For the highest adulterants a decrease in the content of phenols (TPC) by approximately 30% was found and a similar loss for the reducing power (FRAP) was observed. The ability to scavenge DPPH free radical was lowered by as much as over 50%. A similar tendency to decrease antioxidant properties and total phenolic content of honey adulterated with sugar syrups was previously observed, including the dependence of the decrease on the share of sugar syrups [12,41,42,43]. Such a trend could be explained by the dilution effect: introducing a certain increasing volume of syrup into the honey does not introduce new biologically active compounds, it only dilutes the sample. A clear trend was not observed in the distribution of antioxidants between the two phases formed during honey crystallization (Table 2). Since the antioxidant properties of honey are one of the most important features shaping its bioactivity, low antioxidant activity can suggest the potential addition of foreign substances, i.e., sugar syrups. These indicators have been distinguished as useful for the verification of the authenticity of the honey and for the identification of potential misprocessing practices [42].

As the enzymes occurring in honey can be of bee or plant origin, they seem to be good markers of honey adulteration [1]. It has been reported, that one of the ways for adulteration is an addition of the microbiological diastase to syrup. However, our examination indicated that high quality raw honey, exhibiting high initial diastase number, even after addition of 30% of syrup still allows to maintain the diastase number (Table 2) on the demanded level above 8 [28]. Although the diastase activity gradually decreased with both syrup additions, the observed changes were more significant in the case of sugar syrup. We found previously that some glycosidases, such as β-galactosidase (β-GAL) and N-acetyl-β-glucosaminidase (NAG), can be used as sensitive markers of honey quality and thermal processing [44]. In the present study we have found a significant decrease of tested enzymes not exceeding 30%, especially for β-GAL, as the effect of raw honey dilution with syrup (Figure 3). However, the use of those indicators for the detection of adulterations is questionable, as a large variation in enzymatic activity occurs between honey varieties [21,44].

### 2.4. Thermal Analysis

Figure 4 shows the dependence of heat flow rate on temperature for the raw control honey, syrup and invert, which were obtained based on the standard DSC measurements. All analyses were obtained from the second heating scan of the sample at the rate of 10 °C/min, after it has been cooled in a controlled manner at the rate of 10 °C/min. It was found that the adulteration of the control honey with syrup or invert changed the values of the glass transition temperatures. In the upper right corner of the Figure 4, a qualitative analysis of the glass transition was demonstrated for the raw control honey. The inset shows the enlargement of the glass transition area together with temperature of glass transition (T_g_) which was determined at the half height of the jump of heat capacity (ΔC_p_). The change of heat capacity was determined as the height between the extension of liquid and solid heat capacity lines.

The value of the glass transition temperature depends on the chemical structure of the substance, molecular weight, stiffness of the molecular chain or accompanying excipients. The most frequently added substances in honey adulteration are sugar syrups with high fructose contents, invert syrup, rice syrup, maple syrup [6,7,8], as well as sucrose [9] or fructose to glucose [10]. In glass transition region, there is also a change of the physicochemical properties in the material, such as viscosity, density or modulus of elasticity [45]. The control raw honey was tested in the range of −90 °C to 100 °C and two glass transitions were observed for the raw control honey at the temperature of T_g2_ = (−39.5 °C) and T_g5_ = 55.2 °C, respectively (Table 3).

They are probably linked with sucrose, fructose and glucose, but the estimated thermal parameters may also be affected by other ingredients, such as water or proteins. Calorimetric studies [46] show that amorphous fructose undergoes irreversible processes at temperatures above temperature of glass transition, which in the literature are often described as a tautomeric equilibrium [47,48,49,50], anomerization [51] or microheterogeneity [52]. Heat anomalies were observed in the liquid state at the temperature T_1_ = 51 °C and T_2_ = 94 °C [47,52], which have been described as tautomeric processes [47]. Crystalline fructose contains one type of tautomer, β-D-fructopyranose, which after melting mutarrotates to α-fructopyranose, β-D-fructofuranose and α-fructofuranose. The composition of the tautomeric mixture depends on the temperature and the type of solvent used in the study of fructose solutions [47,48]. The liquid state of fructose, which results from melting, is a non-equilibrium state [47,48,49] in which the dominant form is β-D-fructopyranose.

Changes in the thermodynamic properties of amorphous fructose in the glass transition area, which are caused by the influence of different measurement conditions (temperature and time of isothermal annealing) have been presented in the literature [53,54]. The values of the glass transition temperature (T_g_) of fructose are observed in the range of 7 to 17.8 °C [54], and ΔC_p_ was calculated based on changes of the heat flow or the difference between the heat capacity of solid and liquid state at T_g_ is 0.74–0.77 J/(g·°C) [47,53] or 0.88 J/(g·°C) [55]. It can be seen that the glass transition described by the parameters T_g2_ = (−39.50 °C) and ΔC_p2_ = 0.7207 J/(g·°C) is related to the fructose present in the control honey sample. This is expressed by a jump of the heat capacity (ΔC_p2_) resulting from the contents of individual chemical groups—enable for conformation at T_g_.

Moreover, fructose is initially a completely crystalline substance (melting temperature, Tm = 126 °C), which undergoes a complete or partial amorphization after melting. The degradation process occurring in the T_m_ region significantly influences the T_g_ values of fructose in a fully or partially amorphous fructose. The glass transition temperature of the partially degraded fructose sample has the lowest value compared to the T_g_ of the non-degraded or semi-crystalline sample, whose T_g_ reaches the highest value [47,53]. The studies [47,53] clearly show that thermal decomposition of fructose in the melting region reduces the T_g_ value of amorphous or semi-crystalline samples as well as the intensity of tautomeric processes in the amorphous liquid phase.

The honey composition contains a small amount of sucrose. The sucrose content of control honey can be related to the parameters T_g5_ = 55.20 °C and ΔCp5 = 0.1834 J/(g·°C). According to the studies presented in the references [56,57,58,59], the T_g_ value of amorphous sucrose is observed in the range of 52 °C to 75 °C. Large discrepancies for the estimated T_g_ values resulted from measurements carried out with different heating rates, accuracy of analysis, thermal degradation of the sample at T_m_ [57], melting kinetics of crystalline sucrose and the influence of residual water [57]. The influence of the maximum measurement temperature, heating rate and time of annealing a crystalline sucrose sample at a constant temperature on the T_g_ value was presented in [57]. Increasing the final measurement temperature, annealing time or decreasing the heating rate of crystalline sucrose leads to a decrease in the T_g_ value. The observed relationships are related to the increase in the number of molecules with a lower molecular weight (glucose, fructose), caused by the degradation of sucrose. The use of more drastic heating conditions (higher temperature, longer annealing time, very slow heating) increases the T_g_ due to the greater proportion of particles with a high molecular weight, formed in the process of condensation of smaller molecules, which in turn are formed as a result of thermal decomposition of sucrose [57,60]. The existing of glucose in Figure 2 can be additionally indicated by the elongated shape of the glass transition area, what indicates that these two events are derived from sucrose and fructose, since the glass transition temperature of amorphous glucose was determined in the range of 35–42 °C [53].

When both used adulterants were tested in the same way, three glass transitions were observed for the sugar syrup at T_g1_ = (−65.3 °C), T_g2_ = (−40.8 °C), and T_g5_ = 64.5 °C, respectively whereas for the invert glass transitions at temperatures T_g1_ = (−64.0 °C), T_g2_ = (−20.3 °C), and T_g3_ = (−2.55 °C) were found (Table 3). The first glass transitions of syrup and invert show a significantly lower value of T_g1_ than the raw honey control, i.e., −65.3 °C and −64 °C, respectively. The glass transition temperature of the syrup, T_g2_, remains at a similar level of c.a. −40 °C as for the control raw honey. In the case of invert, T_g2_ = (−20.3 °C) was observed. Moreover, an additional slight transition at −2.5 °C was observed for the invert, which was absent both in honey and syrup. The glass transition temperature of the syrup (T_g5_) was observed at 64.5 °C.

Adulterated honey samples show an increase in the glass transition temperature, T_g2_ (Figure 5) for invert (max. 5.7 °C) and a decrease in the T_g2_ value for syrup (max. 10.4 °C) as compared to the control honey (Table 3). In all adulterated honeys, an additional T_g4_ was observed in the range of 34–38.05 °C, which can be related to the presence of glucose and single glass transition apart from sucrose in the honey control sample. Glucose is completely amorphized after it cools down from the melt of its crystalline form [53]. The glass transition of amorphous glucose has been characterized and described in the references [53]. The glass transition temperature of glucose was determined in the range of 35 °C–42 °C which is almost within the range of the observed glass transition at T_g4_. Moreover, in honey with the addition of syrup, a T_g3_ transition occurs in a wide range from −19.5 °C to 4.10 °C and there is no T_g5_ transition present in honey with the addition of invert (50.4–57.6 °C) (Table 3). These specific parameters can be used to identify adulteration of honey with syrup or invert.

## 3. Materials and Methods

### 3.1. Material and Reagents

Multifloral honey was obtained from a local ecological apiary (Jerzy Bańkowski, Więckowice, Poland) in the 2020 season. Two sugar syrups were used in the preparation of the samples: Apikand Premium sugar syrup (Arctos Creme, Bydgoszcz, Poland) and Thymo Invert (BKV Group, Bileća, Bosnia and Herzegovina). The used syrups differed in terms of composition according to the information given on the label (Table 4). All chemicals that were used are of analytical grade. 

### 3.2. Preparation of Model Honey Adulterated with Syrups

Liquid multifloral honey was mixed with each of the two syrups in the proportions of 5, 10, 20 and 30% (*w*/*w*) and allowed to stabilize under room conditions. The original natural honey was used as a control. All samples were stored in room condition for 24 months until analyses.

### 3.3. Physicochemical Parameters

The water content was determined using a Hanna HI96800 (Hanna Instruments, Woonsocket, RI, USA) refractometer dedicated to honey. Measurements were taken at temperature 21 ± 2 °C.

Water activity was measured with the HC2-AW probe and HW5 software (Rotronic AG, Bassersdorf, Switzerland) at 25 ± 2 °C.

The pH was measured in the 20% aqueous honey solutions with Seven Compact pH-meter (Mettler Toledo, Columbus, OH, USA).

The conductivity was determined for the 20% solutions of honey using a CP-401 conductometer (Elmetron, Zabrze, Poland). The results were expressed in mS/cm.

The color of the honey was analyzed using a HI 96785 colorimeter (Hanna Instruments, Woonsocket, RI, USA) dedicated to honey. The results were demonstrated on the Pfund scale.

The viscosity of the honey was analyzed using the me-vi Rotavisc viscosimeter (IKA, Staufen im Breisgau, Germany). Measurements were made at temperature of 18 ± 2 °C with 10,000 rpm spindle speed.

The concentration of HMF was determined colorimetrically strictly according to the procedure described by Dżugan et al. (2021) [21]. Results were expressed as mg of HMF per kg of honey.

Fructose/glucose (F/G) ratio was established based on chromatographic analysis of the sugars in the honey samples using the Camag HPTLC set (Camag, Muttenz, Switzerland). The analysis was carried out strictly according to Tomczyk et al. (2022) [61]. For separation honey solutions were used at concentration of 1 mg/mL. The standards of glucose, fructose and sucrose at a concentration of 0.25 mg/mL were used for identification of sugars in samples. The results were analyzed using VisionCATS 3.2 software (Camag), using the generated chromatograms, the measured peak heights were used to determine the F/G ratio.

### 3.4. Bioactivity Assays

The total phenolic compounds, flavonoids and antioxidant capacity (DPPH and FRAP methods) were determined according to the methodology previously described by Tomczyk et al. (2021) [61]. For the analysis, 20% solutions of honey in distilled water were used.

The protein content in honey was determined using the Bradford method according to Latimer (2016) [62]. Briefly, 1 mL of Bradford reagent (Biorad, Hercules, CA, USA) was added to 100 μL of each honey solution (20% in distilled water). Samples were incubated for 5 min at ambient temperature and the absorbance was read at 595 nm using a spectrophotometer (Biomate 3, Thermo Scientific, Waltham, MA, USA). The results were calculated on the basis of a calibration curve (y = 31.752x, r^2^ = 0.9919) prepared for bovine serum albumin in the range of 6.25–200 µg.

Diastase number was determined by spectrophotometric method with the Phadebas Honey Diastase test (Magle AB, Lund, Sweden) strictly according to the manufacturer′s instructions. The values of the diastase number (DN) were calculated using the following Equation (1):DN = 28.2 × A + 2.64(1)

The activity of three glycosidases: N-acetyl-β-glucosaminidase (NAG), β-galactosidase (β-GAL) and acid phosphatase (AP) was determined in tested honey samples according to the procedure described by Sidor et al. (2021) [44], using appropriate p-nitrophenolic substrate. The absorbance of released *p*-nitrophenol was measured using microplate reader (EPOCH2, BioTek, Winooski, VT, USA) at λ = 400 nm. Results were expressed as enzymatic units U (µmol/min/g).

### 3.5. DSC Analysis

All experiments by calorimetry in the temperature range from −90 °C to 100 °C were performed using the Differential Scanning Calorimeter (DSC) a Q1000TM from TA Instruments, Inc. (New Castle, DE, USA). This calorimeter is the heat-flux type and is equipped with a mechanical refrigerator to control heating and cooling the samples. Measurements were carried out in a nitrogen atmosphere with a constant flow rate of around 50 mL/min. The mass of the samples used for measurements by DSC was ca.10 mg.

The series of experimental heat-flows were obtained at heating rates of 10 °C/min after previous cooling at rate of 10 °C/min. The temperature and heat—flow rate calibration in the DSC apparatus was performed using parameters of melting indium [T_m(onset)_ = 156.6 °C, ΔH_f_ = 28.45 J/g (3.281 kJ/mol)] [45]. In order to obtain accurate results, the heat capacity was calibrated to a sample of sapphire [45]. The accuracy of the measurements was estimated to be ± 1% or better.

### 3.6. Statistical Analysis

For the obtained data, mean values and standard deviations were calculated. Significant differences between different variants of the samples were checked using ANOVA one-way analysis of variance followed by a Tukey’s (HSD) test considering the significance at *p* < 0.05. All measurements were made in triplicate. All calculations were made using Statistica 13.3 software (StatSoft, Tulsa, OK, USA).

## 4. Conclusions

The addition of sugar syrup to honey affected the course of crystallization of artificially adulterated honeys in a syrup type- and dose-dependent manner. During long-term storage, the effect of delamination of products was observed for adulterated honeys which was not observed for raw honey. Along with the increasing share of syrup, adulterated honeys showed a changed chemical composition and reduced biological activity measured as antioxidant and enzymatic activity. This can be explained by the dilution effect of the sample.

Among the tested parameters of artificially adulterated honey, the most evident changes for the viscosity and heat flow rate course obtained by DSC were found. The viscosity of the liquid phase of delaminated honey decreased linearly with the addition of syrup and additional glass transitions specific for used adulterants were found. However, the research was carried out for a single sample of multifloral honey and should be extended to other types of honey. Confirmation of the effectiveness of the proposed tools for identifying adulterated honey, regardless of its variety, would allow the use of both methods in the routine verification of honey authenticity.

## Figures and Tables

**Figure 1 molecules-28-01736-f001:**
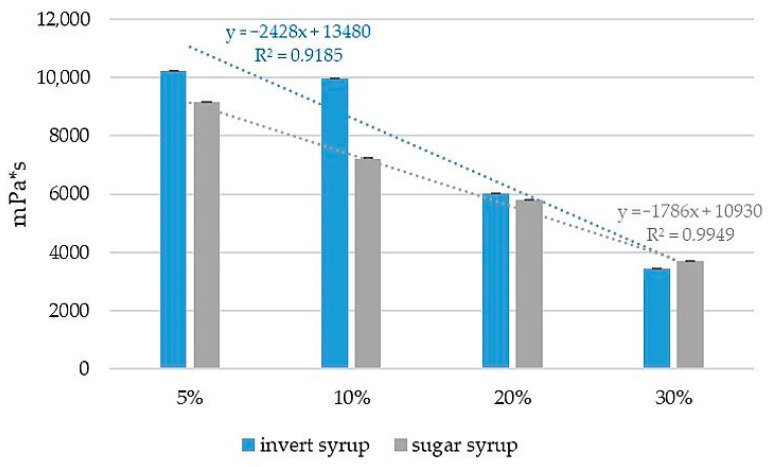
Linear dependence between viscosity and content of sugar syrup in liquid phase of adulterated honey.

**Figure 2 molecules-28-01736-f002:**
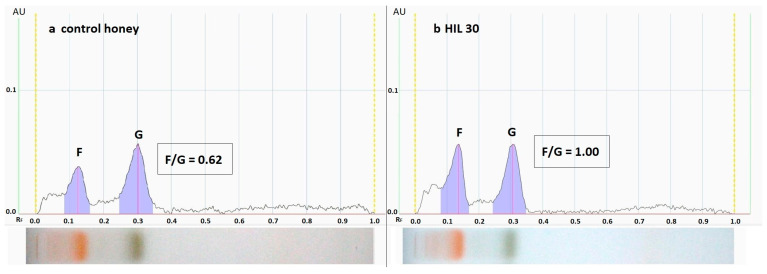
Exemplary of sugar HPTLC chromatograms and generated profiles: (**a**)—control honey, (**b**)—HIL 30, peaks: F—fructose, G—glucose.

**Figure 3 molecules-28-01736-f003:**
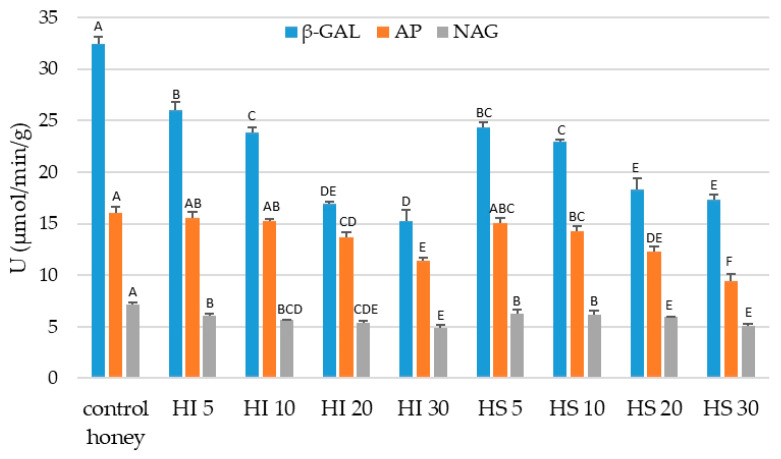
The effect of adulterant addition on the activity of selected native hydrolytic enzymes of honey. ^A,B,C,D,E,F^—samples marked with different superscript letters are statistically significant different within the control honey and adulterated samples (mix only) (separately within individual enzymes) (Tukey’s honest significant difference test, *p* < 0.05).

**Figure 4 molecules-28-01736-f004:**
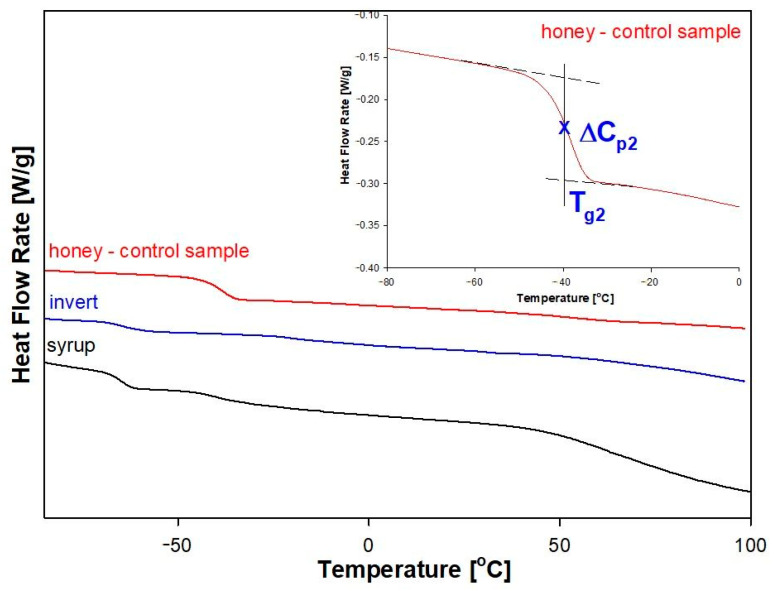
Comparison of heat flow rates of control honey sample (red), syrup (black) and invert (blue) versus temperature. The inset shows the enlargement of the glass transition region together with temperature of glass transition (T_g_) which was determined at the half height of the jump of heat capacity (ΔC_p_) of the control honey sample.

**Figure 5 molecules-28-01736-f005:**
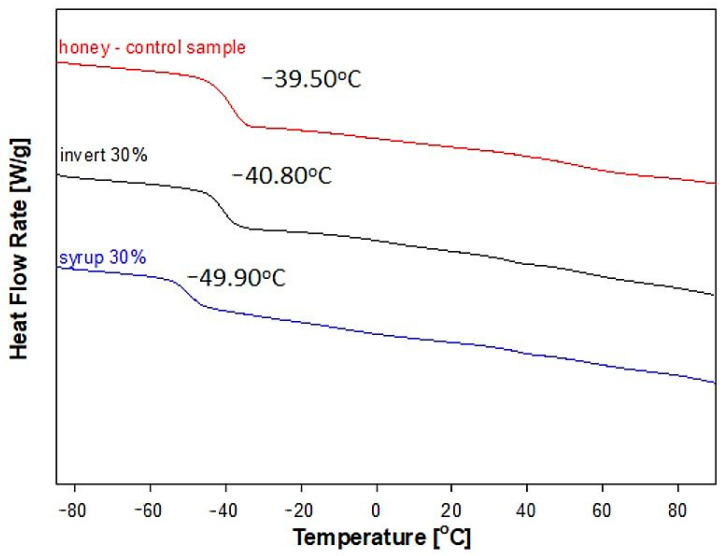
Comparison of heat flow rates of honey control sample and artificially adulterated honey by syrup 30% and invert 30% versus temperature.

**Table 1 molecules-28-01736-t001:** Physicochemical parameters of adulterated honey samples.

Sample/Adulterant		Sample Code	L/S Layer Ratio [*w*/*w*]	Color [Pfund]	Water Content [%]	Water Activity [a_w_]	pH	Conductivity [mS/cm]	Viscosity [mPa*s]	F/G Ratio	HMF [mg/kg]
**Control honey**		H	-	102 ± 1.4 ^A^	17.90 ± 0.1 ^A^	0.569 ± 0.00 ^A^	4.45 ± 0.01 ^A^	0.764 ± 0.014 ^A^	10910 ± 14 ^A^	0.62	5.03 ± 0.29 ^A^
**Invert syrup**		I	-	42 ± 1.4 *	28.9 ± 0.1 *	0.713 ± 0.00 *	5.61 ± 0.03 *	0.171 ± 0.001 *	365 ± 7 *	0.60	10.57 ± 0.36 *
**Sugar syrup**		S	-	6 ± 0.7 *	29.9 ± 0.0 *	0.731 ± 0.00 *	4.78 ± 0.01 *	0.056 ± 0.000 *	355 ± 7 *	0.66	0.00 ± 0.06 *
**Invert addition**							**Invert adulterated honey**				
5%	Mix	HI 5	1	98 ± 0.7 ^B^	19.5 ± 0.1 ^CDa^	0.591 ± 0.00 ^B^	4.64 ± 0.01 ^CE^	0.751 ± 0.001 ^ABa^	n.t.	0.63	14.19 ± 0.52 ^BCa^
	Liquid	HIL 5		99 ± 0.7	19.8 ± 0.1 ^b^	0.589 ± 0.00 ^b^	4.60 ± 0.03	0.774 ± 0.003 ^a^	10215 ± 7 ^B^	0.70	16.72 ± 0.41 ^b^
	Solid	HIS 5		96 ± 0.0	18.2 ± 0.3 ^a^	0.591 ± 0.00 ^ab^	4.66 ± 0.04	0.703 ± 0.003 ^b^	n.t.	0.61	12.23 ± 0.56 ^a^
10%	Mix	HI 10	1.70	95 ± 0.7 ^BC^	19.8 ± 0.1 ^BCa^	0.596 ± 0.00 ^C^	4.74 ± 0.02 ^BD^	0.716 ± 0.002 ^Ca^	n.t.	0.66	15.83 ± 0.31 ^BEa^
	Liquid	HIL 10		96 ± 0.7	20.9 ± 0.2 ^b^	0.596 ± 0.00	4.69 ± 0.02	0.744 ± 0.003 ^b^	9955 ± 7 ^C^	0.75	18.30 ± 0.54 ^b^
	Solid	HIS 10		94 ± 0.7	18.9 ± 0.4 ^c^	0.597 ± 0.00	4.77 ± 0.03	0.649 ± 0.003 ^c^	n.t.	0.55	11.78 ± 0.42 ^c^
20%	Mix	HI 20	2.22	93 ± 0.0 ^Cab^	20.1 ± 0.2 ^BEa^	0.606 ± 0.00 ^D^	4.82 ± 0.05 ^B^	0.676 ± 0.002 ^Da^	n.t.	0.67	17.92 ± 0.47 ^FGa^
	Liquid	HIL 20		94 ± 0.0 ^a^	21.9 ± 0.1 ^b^	0.606 ± 0.00	4.79 ± 0.01	0.702 ± 0.003 ^b^	6030 ± 14 ^D^	0.73	20.24 ± 0.74 ^b^
	Solid	HIS 20		91 ± 0.7 ^b^	19.9 ± 0.1 ^a^	0.607 ± 0.00	4.87 ± 0.02	0.586 ± 0.001 ^c^	n.t.	0.49	16.22 ± 0.34 ^a^
30%	Mix	HI 30	2.70	89 ± 0.7 ^Dab^	21.7 ± 0.1 ^Fa^	0.632 ± 0.00 ^E^	5.07 ± 0.02 ^F^	0.622 ± 0.005 ^Ea^	n.t.	0.81	22.11 ± 0.66 ^H^
	Liquid	HIL 30		91 ± 0.0 ^a^	23.2 ± 0.2 ^b^	0.625 ± 0.00	4.97 ± 0.02	0.687 ± 0.001 ^b^	3445 ± 7 ^E^	1.00	22.98 ± 0.51
	Solid	HIS 30		87 ± 0.7 ^b^	21.4 ± 0.1 ^a^	0.639 ± 0.00	5.14 ± 0.02	0.500 ± 0.001 ^c^	n.t.	0.52	20.99 ± 0.71
**Syrup addition**							**Syrup adulterated honey**				
5%	Mix	HS 5	1.17	96 ± 0.0 ^BC^	19.1 ± 0.2 ^D^	0.587 ± 0.00 ^F^	4.55 ± 0.01 ^C^	0.746 ± 0.00 ^ABa^	n.t.	0.81	12.18 ± 0.41 ^Dab^
	Liquid	HSL 5		97 ± 0.7	20.1 ± 0.7	0.586 ± 0.00	4.53 ± 0.01	0.785 ± 0.002 ^b^	9165 ± 7 ^F^	0.92	13.74 ± 0.88 ^a^
	Solid	HSS 5		94 ± 1.4	18.9 ± 0.2	0.588 ± 0.00	4.57 ± 0.01	0.699 ± 0.000 ^c^	n.t.	0.65	11.18 ± 0.59 ^b^
10%	Mix	HS 10	1.94	87 ± 0.7 ^DEa^	19.5 ± 0.1 ^CD^	0.605 ± 0.00 ^Da^	4.59 ± 0.01 ^C^	0.701 ± 0.001 ^Ca^	n.t.	0.75	13.83 ± 0.56 ^CDa^
	Liquid	HSL 10		89 ± 0.0 ^a^	20.7 ± 0.2	0.601 ± 0.00 ^b^	4.56 ± 0.01	0.725 ± 0.002 ^b^	7230 ± 14 ^G^	0.85	15.23 ± 0.33 ^b^
	Solid	HSS 10		83 ± 0.7 ^b^	19.3 ± 0.2	0.609 ± 0.00 ^c^	4.61 ± 0.01	0.631 ± 0.001 ^c^	n.t.	0.59	10.01 ± 0.47 ^a^
20%	Mix	HS 20	2.84	84 ± 0.7 ^Ea^	20.8 ± 0.2 ^EGab^	0.613 ± 0.00 ^Ga^	4.64 ± 0.02 ^CEab^	0.656 ± 0.002 ^Da^	n.t.	0.74	16.20 ± 0.54 ^EFa^
	Liquid	HSL 20		85 ± 0.0 ^a^	21.5 ± 0.3 ^a^	0.611 ± 0.00 ^a^	4.57 ± 0.00 ^a^	0.688 ± 0.001 ^b^	5805 ± 7 ^D^	0.87	18.97 ± 0.78 ^b^
	Solid	HSS 20		79 ± 0.0 ^b^	19.9 ± 0.1 ^b^	0.615 ± 0.00 ^b^	4.66 ± 0.01 ^b^	0.564 ± 0.001 ^c^	n.t.	0.57	11.28 ± 0.69 ^c^
30%	Mix	HS 30	3.54	78 ± 0.7 ^Fa^	21.3 ± 0.2 ^FG^	0.631 ± 0.00 ^E^	4.70 ± 0.02 ^DEab^	0.563 ± 0.003 ^Fa^	n.t.	0.74	19.01 ± 0.34 ^Ga^
	Liquid	HSL 30		80 ± 0.0 ^a^	22.7 ± 0.5	0.629 ± 0.0	4.67 ± 0.00 ^a^	0.599 ± 0.002 ^b^	3675 ± 7	0.77	21.20 ± 0.50 ^b^
	Solid	HSS 30		75 ± 0.7 ^b^	20.8 ± 0.1	0.631 ± 0.0	4.74 ± 0.02 ^b^	0.473 ± 0.001 ^c^	n.t.	0.51	13.48 ± 0.40 ^c^

n.t.—not tested. *—samples marked with a star are statistically significant different from the control honey (in the column) (Tukey’s honest significant difference test, *p* < 0.05). ^A,B,C,D,E,F,G,H^—means marked with different superscript letters within the column are statistically significant different within the control honey and adulterated samples (mix only) (Tukey’s honest significant difference test, *p* < 0.05). a,b,c—means marked with different superscript letters within the column are statistically significant different within the single sample (mixed, liquid, solid) (Tukey’s honest significant difference test, *p* < 0.05).

**Table 2 molecules-28-01736-t002:** Antioxidant activity, protein content and diastase activity of adulterated honey samples.

Sample	DPPH [µmol TE/100 g]	FRAP [µmol TE/100 g]	TPC [mg GAE/100 g]	Protein Content [mg/100 g]	Diastase [DN]
Control honey	56.82 ± 1.22 ^A^	570.39 ± 6.51 ^A^	229.91 ± 1.05 ^A^	117.95 ± 0.45 ^A^	23.41 ± 0.22 ^A^
Invert syrup	1.81 ± 0.68 *	50.99 ± 0.47 *	55.80 ± 2.31 *	0.94 ± 0.14 *	<0.9 **
Sugar syrup	14.51 ± 2.06 *	112.83 ± 6.05 *	38.99 ± 1.26 *	0.05 ± 0.13 *	<0.9 **
**Invert addition**					
HI 5	47.87 ± 0.67 ^Ba^	508.55 ± 14.89 ^B^	205.36 ± 2.10 ^B^	106.53 ± 1.22 ^B^	22.90 ± 0.06 ^AB^
HIL 5	54.42 ± 0.72 ^b^	518.42 ± 6.51	203.87 ± 1.68	107.16 ± 0.56	20.04 ± 3.91
HIS 5	45.82 ± 0.88 ^a^	493.75 ± 18.14	208.63 ± 4.63	105.43 ± 1.00	21.42 ± 0.48
HI 10	41.85 ± 0.42 ^C^	452.96 ± 2.33 ^CE^	198.21 ± 1.26 ^BC^	104.64 ± 0.78 ^BC^	21.66 ± 0.70 ^C^
HIL 10	43.36 ± 0.86	441.45 ± 5.58	193.15 ± 0.84	102.04 ± 0.22	19.19 ± 0.24
HIS 10	41.85 ± 0.42	442.11 ± 1.86	188.99 ± 2.53	102.04 ± 0.89	19.79 ± 0.28
HI 20	38.61 ± 1.64 ^CD^	426.32 ± 0.93 ^DE^	179.46 ± 2.53 ^D^	101.41 ± 1.11 ^Ca^	15.74 ± 0.22 ^D^
HIL 20	39.19 ± 0.83	415.13 ± 5.58	185.12 ± 1.26	101.80 ± 1.00 ^a^	16.29 ± 0.12
HIS 20	43.23 ± 3.22	421.05 ± 0.93	172.92 ± 2.95	95.82 ± 0.56 ^b^	14.95 ± 0.18
HI 30	34.35 ± 0.39 ^DE^	394.74 ± 6.51 ^DFa^	152.98 ± 2.10 ^Ea^	95.03 ± 1.45 ^D^	10.63 ± 0.22 ^E^
HIL 30	32.56 ± 1.35	453.62 ± 0.47 ^b^	151.79 ± 5.05 ^a^	98.66 ± 1.22	9.58 ± 0.44
HIS 30	36.03 ± 1.20	366.45 ± 1.86 ^a^	165.33 ± 5.26 ^b^	98.26 ± 1.34	9.31 ± 0.18
**Syrup addition**					
HS 5	43.84 ± 3.24 ^C^	511.84 ± 14.89 ^B^	201.64 ± 1.47 ^BCa^	111.65 ± 0.67 ^Eb^	22.86 ± 0.08 ^ABa^
HSL 5	39.78 ± 0.83	525.33 ± 3.26	196.88 ± 1.89 ^b^	106.76 ± 0.89 ^a^	22.79 ± 0.06 ^ab^
HSS 5	47.11 ± 3.57	495.39 ± 3.72	183.48 ± 1.89 ^a^	107.32 ± 1.22 ^a^	22.13 ± 0.24 ^b^
HS 10	38.90 ± 0.41 ^CD^	481.25 ± 6.98 ^BCa^	197.02 ± 3.79 ^Ca^	102.75 ± 0.78 ^BC^	21.51 ± 0.36 ^Ca^
HSL 10	41.70 ± 0.21	496.38 ± 5.12 ^a^	194.05 ± 0.42 ^b^	105.98 ± 0.89	21.17 ± 0.32 ^a^
HSS 10	37.47 ± 2.43	424.01 ± 10.70 ^b^	180.80 ± 2.31 ^a^	102.12 ± 0.11	19.66 ± 0.14 ^b^
HS 20	36.17 ± 1.00 ^D^	444.41 ± 3.26 ^Ea^	186.76 ± 1.47 ^D^	87.63 ± 0.11 ^Fab^	18.90 ± 0.22 ^Fa^
HSL 20	35.47 ± 1.19	474.67 ± 1.40 ^a^	181.55 ± 1.26	89.36 ± 0.56 ^a^	17.29 ± 0.10 ^a^
HSS 20	36.89 ± 1.61	408.88 ± 1.40 ^b^	179.17 ± 2.10	83.93 ± 1.11 ^b^	15.82 ± 0.06 ^b^
HS 30	29.60 ± 0.94 ^E^	373.03 ± 13.03 ^Fa^	168.01 ± 1.89 ^Fa^	78.26 ± 2.67 ^G^	15.98 ± 0.12 ^Da^
HSL 30	32.69 ± 0.39	417.11 ± 8.37 ^b^	161.01 ± 0.42 ^b^	81.96 ± 1.00	15.58 ± 0.12 ^a^
HSS 30	28.94 ± 0.37	356.25 ± 17.21 ^a^	142.11 ± 2.74 ^a^	82.51 ± 1.11	14.79 ± 0.16 ^b^

*—samples marked with a star are statistically significant different from control honey (in the column) (Tukey’s honest significant difference test, *p* < 0.05). **—below detection limit in Phadebas test. ^A,B,C,D,E,F^—means marked with different superscript letters within the column are statistically significant different within the control honey and adulterated samples (mix only) (Tukey’s honest significant difference test, *p* < 0.05). ^a,b,c^—means marked with different superscript letters within the column are statistically significant different within the single sample (mix, liquid, solid) (Tukey’s honest significant difference test, *p* < 0.05).

**Table 3 molecules-28-01736-t003:** Comparison of thermal parameters of control honey, syrup, invert, and adulterated honey samples (mean ± SD).

Sample	T_g1_ [°C]	ΔC_p1_ [J/(g·°C)]	T_g2_ [°C]	ΔC_p2_ [J/(g·°C)]	T_g3_ [°C]	ΔC_p3_ [J/(g·°C)]	T_g4_ [°C]	ΔC_p4_ [J/(g·°C)]	T_g5_ [°C]	ΔC_p5_ [J/(g·°C)]
Control Honey	-	-	−39.50 ± 0.05	0.7207 ± 0.0073	-	-	-	-	55.20 ± 0.05	0.1834 ± 0.0018
Invert Syrup	−64.00 ± 0.05	0.2687 ± 0.0027	−20.30 ± 0.05	0.1198 ± 0.0012	−2.55 ± 0.05	0.0443 ± 0.0044	-	-	-	-
Sugar Syrup	−65.30 ± 0.05	0.5087 ± 0.0051	−40.80 ± 0.05	0.2428 ± 0.0024	-	-	-	-	64.50 ± 0.05	1.4600 ± 0.0150
HI 5	-	-	−34.60 ± 0.05	0.5781 ± 0.0058	-	-	38.05 ± 0.05	0.0706 ± 0.0071	50.40 ± 0.05	0.0064 ± 0.0006
HI 10	-	-	−35.40 ± 0.05	0.7229 ± 0.0072	-	-	35.20 ± 0.05	0.0018 ± 0.0001	57.60 ± 0.05	0.1315 ± 0.0013
HI 20	-	-	−33.80 ± 0.05	0.5854 ± 0.0059	-	-	36.90 ± 0.05	0.0931 ± 0.0009	57.50 ± 0.05	0.1234 ± 0.0013
HI 30	-	-	−40.80 ± 0.05	0.5725 ± 0.0057	-	-	34.50 ± 0.05	0.0765 ± 0.0008	56.40 ± 0.05	0.1347 ± 0.0013
HS 5	-	-	−45.60 ± 0.05	0.3189 ± 0.0032	4.10 ± 0.05	0.1947 ± 0.0019	34.00 ± 0.05	0.0185 ± 0.0019	-	-
HS 10	-	-	−42.40 ± 0.05	0.4509 ± 0.0045	−6.40 ± 0.05	0.2028 ± 0.0020	36.50 ± 0.05	0.0576 ± 0.0006	-	-
HS 20	-	-	−42.20 ± 0.05	0.3744 ± 0.0037	−19.50 ± 0.05	0.1367 ± 0.0014	36.00 ± 0.05	0.0534 ± 0.0006	-	-
HS 30	-	-	−49.90 ± 0.05	0.4294 ± 0.0043	−9.80 ± 0.05	0.1357 ± 0.0014	36.70 ± 0.5	0.0773 ± 0.0077	-	-

**Table 4 molecules-28-01736-t004:** Characteristics of used adulterants.

**Adulterant**	**Sugar Syrup**	**Invert**
	Apikand Premium sugar syrup (Arctos Creme, Bydgoszcz, Poland)	Thymo Invert (BKV Group, Bileća, Bosnia and Herzegovina)
**Composition according to the manufacturer information on the label**	Glucose: 37% Fructose: 33.5% Sucrose: 29.5% Water	Sugar (sucrose, glucose, fructose): 70% Plant extracts and essential oils: 0.33% Water: 29.67%

## Data Availability

The data presented in this study are available in the article.

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
