# Peer review of "Tracking of Thermal, Physicochemical, and Biological Parameters of a Long-Term Stored Honey Artificially Adulterated with Sugar Syrups"

_molecules, 2023, doi:10.3390/molecules28041736_

Round 1

Reviewer 1 Report

The current study is a good presence in the scientific soundness and knowledge. However, some points need to revise and make the clarify from authors.

- First, almost the material and method section detected high plagiarism, the author needs to re-write it. The evidence was attached PDF file herewith. Please check it out. 

- In figure 3, the result is not clear for the invert line compared to the control and syrup treatment. Please justify this point.

- What is the reason behind of the study for determining the conductivity of solution honey.?

- In antioxidant activity, whether the Invert test and syrup test do seem not different results, so how this treatment was vital for the long-term storage of the sample solution? It is unclear and needs clarification

- Check the abbreviations throughout the manuscript and introduce the abbreviation when the full word appears the first time in the abstract and the remaining for the text and then use only the abbreviation. Also, the typing error needs to check.

- In the conclusion seems to be in general and is not given separately, it is highly recommended to include limitations of the study and potential future research goals.

Author Response

Dear Reviewer,

thank you for your comments and valuable suggestions, which allowed to improve our manuscript. All editorial and language recommendations have been incorporated into the text, moreover, the work has been linguistically checked by an English lector. Below we send details of the changes introduced in the manuscript, as well as explanations for your doubts or inaccuracies.

Sincerely,

Authors

Reviewer 2 Report

Authors report a thorough evaluation of physical and chemical properties of pristine honey  vs. adulterated honey. Although this is valuable information I believe a more clear message needs to be reported by the authors. I have difficulty to follow the long descriptive narratives of the results. Terms as "slightly high" should be avoided as they are subjective. 

Many of the variables shown do not present evident differences although they are reported as significantly different. Other don't show statistical treatment.

I believe the text needs to report in a - succinct - and clear manner the results and would advise English revision. In some instances punctuation is missing and it is difficult to follow what the authors are trying to convey.

I have difficulty understanding the abstract "The adulterated samples were characterized by an elevated values of physicochemical  parameters and lower bioactivity" - Should be "by elevated values of ...."

Physicochemical parameters is vague. I don't know what it means. Low bioactivity I also am not sure what it means. The abstract should give a clear message of the finidings.

"Using the differential scanning  calorimetry (DSC) specific glass transition"  - please put a comma after  (DSC)

"Moreover, the additional different 23 glass transitions for adulterated honeys: -19.5°C to 4.10°C (in the case of syrup) and 50.4 to 57.6°C 24 (for invert) were noticed."  -  I don't understand "were noticed" - Why is this important in comparison the non-adulterated control samples?

Results and discussion:

The graphic presented show no difference between control and HI30. What about all the other HI? "HS 5, HS 83 10, HS 20"? no chart comparison? HI30 is really high so one assumes the others would also be not significant? but we cannot assume.

Table 1 seems to indicate very little differences between control, invert syrup and sugar addition except for conductivity, viscosity and HMF. Ph 4.45. and Ph 4.78 are statistically different?  Viscosity 10910 and viscosity 355 no statisticlly significant?

Table 2 is informative and shows evident differences.

Table 3 does not show statistical significances

Author Response

(The authors gave the same response as above.)

Reviewer 3 Report

The article “Tracking of thermal, physicochemical, and biological parameters of a long-term stored honey artificially adulterated with sugar syrups”:In this study, The addition of sugar syrup to honey affected the course of crystallization of artificially adulterated honeys in a syrup type- and dose-dependent manner. Along with the increasing share of syrup, adulterated honeys showed a changed chemical composition and reduced biological activity measured as antioxidant and enzymatic activity. 

1. result 2.1 mentions that “More glucose results in faster crystallization of honey”.  the validity of pH as an adulterant indicator is therefore questionable

2. The typical range for honey is 3.4-6.1, and the ph of the adulterated honey sample is within the standard range. Whether the effect of pH level can be used as an evaluation criterion.

3. result 2.2 mentions that “the measurements of viscosity may serve as an indicator of the presence of adulterants.” The author does not provide charts, so it is recommended to add.

4. There are many types of honey, and whether they all conform to the experimental phenomena derived in this manuscript requires additional sample testing to support the generality of the application of the method.

Author Response

(The authors gave the same response as above.)

Round 2

Reviewer 2 Report

Thank you for the efforts to make the information more clear and succinct.

Reviewer 3 Report

The article “Tracking of thermal, physicochemical, and biological parameters of a long-term stored honey artificially adulterated with sugar syrups”:In this study, The addition of sugar syrup to honey affected the course of crystallization of artificially adulterated honeys in a syrup type- and dose-dependent manner. Along with the increasing share of syrup, adulterated honeys showed a changed chemical composition and reduced biological activity measured as antioxidant and enzymatic activity. The article has been revised according to the opinions. After revision, the readability of the article has been improved.